# DECOUPLING REPRESENTATION LEARNING FROM REINFORCEMENT LEARNING

## ABSTRACT

In an effort to overcome limitations of reward-driven feature learning in deep reinforcement learning (RL) from images, we propose decoupling representation learning from policy learning. To this end, we introduce a new unsupervised learning (UL) task, called Augmented Temporal Contrast (ATC), which trains a convolutional encoder to associate pairs of observations separated by a short time difference, under image augmentations and using a contrastive loss. In online RL experiments, we show that training the encoder exclusively using ATC matches or outperforms end-to-end RL in most environments. Additionally, we benchmark several leading UL algorithms by pre-training encoders on expert demonstrations and using them, with weights frozen, in RL agents; we find that agents using ATC-trained encoders outperform all others. We also train multi-task encoders on data from multiple environments and show generalization to different downstream RL tasks. Finally, we ablate components of ATC, and introduce a new data augmentation to enable replay of (compressed) latent images from pre-trained encoders when RL requires augmentation. Our experiments span visually diverse RL benchmarks in DeepMind Control, DeepMind Lab, and Atari, and our complete code is available at `hiddenurl`.

## 1 INTRODUCTION

Ever since the first fully-learned approach succeeded at playing Atari games from screen images (Mnih et al., 2015), standard practice in deep reinforcement learning (RL) has been to learn visual features and a control policy jointly, end-to-end. Several such deep RL algorithms have matured (Hessel et al., 2018; Schulman et al., 2017; Mnih et al., 2016; Haarnoja et al., 2018) and have been successfully applied to domains ranging from real-world (Levine et al., 2016; Kalashnikov et al., 2018) and simulated robotics (Lee et al., 2019; Laskin et al., 2020a; Hafner et al., 2020) to sophisticated video games (Berner et al., 2019; Jaderberg et al., 2019), and even high-fidelity driving simulators (Dosovitskiy et al., 2017). While the simplicity of end-to-end methods is appealing, relying on the reward function to learn visual features can be severely limiting. For example, it leaves features difficult to acquire under sparse rewards, and it can narrow their utility to a single task. Although our intent is broader than to focus on either sparse-reward or multi-task settings, they arise naturally in our studies. We investigate how to learn visual representations which are agnostic to rewards, without degrading the control policy.

A number of recent works have significantly improved RL performance by introducing auxiliary losses, which are unsupervised tasks that provide feature-learning signal to the convolution neural network (CNN) encoder, additionally to the RL loss (Jaderberg et al., 2017; van den Oord et al., 2018; Laskin et al., 2020b; Guo et al., 2020; Schwarzer et al., 2020). Meanwhile, in the field of computer vision, recent efforts in unsupervised and self-supervised learning (Chen et al., 2020; Grill et al., 2020; He et al., 2019) have demonstrated that powerful feature extractors can be learned without labels, as evidenced by their usefulness for downstream tasks such as ImageNet classification. Together, these advances suggest that visual features for RL could possibly be learned entirely without rewards, which would grant greater flexibility to improve overall learning performance. To our knowledge, however, no single unsupervised learning (UL) task has been shown adequate for this purpose in general vision-based environments.

In this paper, we demonstrate the first decoupling of representation learning from reinforcement learning that performs as well as or better than end-to-end RL. We update the encoder weights using only UL and train a control policy independently, on the (compressed) latent images. This capability stands in contrast to previous state-of-the-art methods, which have trained the UL and RL objectives jointly, or Laskin et al. (2020b), which observed diminished performance with decoupled encoders.

Our main enabling contribution is a new unsupervised task tailored to reinforcement learning, which we call Augmented Temporal Contrast (ATC). ATC requires a model to associate observations from nearby time steps within the same trajectory (Anand et al., 2019). Observations are encoded via a convolutional neural network (shared with the RL agent) into a small latent space, where the InfoNCE loss is applied (van den Oord et al., 2018). Within each randomly sampled training batch, the positive observation, $o_{t+k}$, for every anchor, $o_t$, serves as negative for all other anchors. For regularization, observations undergo stochastic data augmentation (Laskin et al., 2020b) prior to encoding, namely random shift (Kostrikov et al., 2020), and a momentum encoder (He et al., 2020; Laskin et al., 2020b) is used to process the positives. A learned predictor layer further processes the anchor code (Grill et al., 2020; Chen et al., 2020) prior to contrasting. In summary, our algorithm is a novel combination of elements that enables generic learning of the structure of observations and transitions in MDPs without requiring rewards or actions as input.

We include extensive experimental studies establishing the effectiveness of our algorithm in a visually diverse range of common RL environments: DeepMind Control Suite (DMControl; Tassa et al. 2018), DeepMind Lab (DMLab; Beattie et al. 2016), and Atari (Bellemare et al., 2013). Our experiments span discrete and continuous control, 2D and 3D visuals, and both on-policy and off policy RL algorithms. Complete code for all of our experiments is available at `hiddenurl`. Our empirical contributions are summarized as follows:

*Online RL with UL*: We find that the convolutional encoder trained solely with the unsupervised ATC objective can fully replace the end-to-end RL encoder without degrading policy performance. ATC achieves nearly equal or greater performance in all DMControl and DMLab environments tested and in 5 of the 8 Atari games tested. In the other 3 Atari games, using ATC as an auxiliary loss or for weight initialization still brings improvements over end-to-end RL.

*Encoder Pre-Training Benchmarks*: We pre-train the convolutional encoder to convergence on expert demonstrations, and evaluate it by training an RL agent using the encoder with weights frozen. We find that ATC matches or outperforms all prior UL algorithms as tested across all domains, demonstrating that ATC is a state-of-the-art UL algorithm for RL.

*Multi-Task Encoders*: An encoder is trained on demonstrations from multiple environments, and is evaluated, with weights frozen, in separate downstream RL agents. A single encoder trained on four DMControl environments generalizes successfully, performing equal or better than end-to-end RL in four held-out environments. Similar attempts to generalize across eight diverse Atari games result in mixed performance, confirming some limited feature sharing among games.

*Ablations and Encoder Analysis*: Components of ATC are ablated, showing their individual effects. Additionally, data augmentation is shown to be necessary in DMControl during RL even when using a frozen encoder. We introduce a new augmentation, *subpixel random shift*, which matches performance while augmenting the latent images, unlocking computation and memory benefits.

## 2  RELATED WORK

Several recent works have used unsupervised/self-supervised representation learning methods to improve performance in RL. The UNREAL agent (Jaderberg et al., 2017) introduced unsupervised auxiliary tasks to deep RL, including the Pixel Control task, a Q-learning method requiring predictions of screen changes in discrete control environments, which has become a standard in DMLab (Hessel et al., 2019). CPC (van den Oord et al., 2018) applied contrastive losses over multiple time steps as an auxiliary task for the convolutional and recurrent layers of RL agents, and it has been extended with future action-conditioning (Guo et al., 2018). Recently, PBL (Guo et al., 2020) surpassed these methods with an auxiliary loss of forward and backward predictions in the recurrent latent space using partial agent histories. Where the trend is of increasing sophistication in auxiliary recurrent architectures, our algorithm is markedly simpler, requiring only observations, and yet it proves sufficient in partially observed settings (POMDPs).

ST-DIM (Anand et al., 2019) introduced various temporal, contrastive losses, including ones that operate on "local" features from an intermediate layer within the encoder, without data augmentation. CURL (Laskin et al., 2020b) introduced an augmented, contrastive auxiliary task similar to ours, including a momentum encoder but without temporal contrast. Mazoure et al. (2020) provided extensive analysis pertaining to InfoNCE losses on functions of successive time steps in MDPs, including local features in their auxiliary loss (DRIML) similar to ST-DIM, and finally conducted experiments using global temporal contrast of augmented observations in the Procgen (Cobbe et al., 2019) environment. Most recently, MPR (Schwarzer et al., 2020) combined data augmentation with multi-step, convolutional forward modeling and a similarity loss to improve DQN agents in the Atari 100k benchmark. Hafner et al. (2019; 2020); Lee et al. (2019) proposed to leverage world-modeling in a latent-space for continuous control. A small number of model-free methods have attempted to decouple encoder training from the RL loss as ablations, but have met reduced performance relative to end-to-end RL (Laskin et al., 2020b; Lee et al., 2020). None have previously been shown effective in as diverse a collection of RL environments as ours (Bellemare et al., 2013; Tassa et al., 2018; Beattie et al., 2016).

Finn et al. (2016); Ha & Schmidhuber (2018) are example works which pretrained encoder features in advance using image reconstruction losses such as the VAE (Kingma & Welling, 2013). Devin et al. (2018); Kipf et al. (2019) pretrained object-centric representations, the latter learning a forward model by way of contrastive losses; Yan et al. (2020) introduced a similar technique to learn encoders supporting manipulation of deformable objects by traditional control methods. MERLIN (Wayne et al., 2018) trained a convolutional encoder and sophisticated memory module online, detached from the RL agent, which learned read-only accesses to memory. It used reconstruction and one-step latent-prediction losses and achieved high performance in DMLab-like environments with extreme partial observability. Our loss function may benefit those settings, as it outperforms similar reconstruction losses in our experiments. Decoupling unsupervised pretraining from downstream tasks is common in computer vision (Hénaff et al., 2019; He et al., 2019; Chen et al., 2020) and has favorable properties of providing task agnostic features which can be used for training smaller task-specific networks, yielding significant gains in computational efficiency over end-to-end methods.

# 3 AUGMENTED TEMPORAL CONTRAST

Our unsupervised learning task, Augmented Temporal Contrast (ATC), requires a model to associate an observation, $o_t$, with one from a specified, near-future time step, $o_{t+k}$. Within each training batch, we apply stochastic data augmentation to the observations (Laskin et al., 2020b), namely random shift (Kostrikov et al., 2020), which is simple to implement and provides highly effective regularization in most cases. The augmented observations are encoded into a small latent space where a contrastive loss is applied. This task encourages the learned encoder to extract meaningful elements of the structure of the MDP from observations.

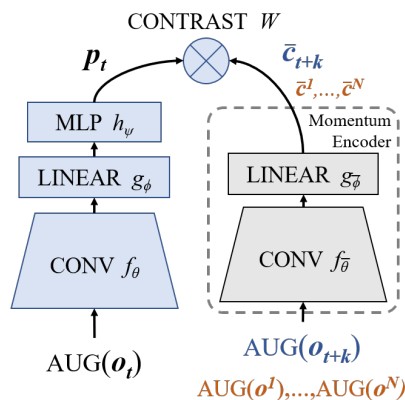

Our architecture for ATC consists of four learned components - (i) a convolutional *encoder*, $f_\theta$, which processes the anchor observation, $o_t$, into the latent image $z_t = f_\theta(\text{AUG}(o_t))$, (ii) a linear *global compressor*, $g_\phi$ to produce a small latent code vector $c_t = g_\phi(z_t)$, (iii) a residual *predictor* MLP, $h_\psi$, which acts as an implicit forward model to advance the code $p_t = h_\psi(c_t) + c_t$, and (iv) a *contrastive transformation* matrix, $W$. To process the positive observation, $o_{t+k}$ into the target code $\bar{c}_{t+k} = g_{\bar\phi}(f_{\bar\theta}(\text{AUG}(o_{t+k}))$, we use a momentum encoder (He et al., 2019) parameterized as a slowly moving average of the weights from the learned encoder and compressor layer:

Figure 1: Augmented Temporal Contrast— augmented observations are processed through a learned encoder $f_\theta$, compressor, $g_\phi$ and residual predictor $h_\psi$, and are associated through a contrastive loss with a positive example from $k$ time steps later, processed through a momentum encoder.

$$\bar\theta \leftarrow (1-\tau)\bar\theta + \tau\theta \; ; \qquad \bar\phi \leftarrow (1-\tau)\bar\phi + \tau\phi \,. \tag{1}$$

The complete architecture is shown in Figure 1. The convolutional encoder, $f_\theta$, alone is shared with the RL agent.

We employ the InfoNCE loss (Gutmann & Hyvärinen, 2010; van den Oord et al., 2018) using logits computed bilinearly, as $l = p_t W \bar{c}_{t+k}$. In our implementation, every anchor in the training batch utilizes the positives corresponding to all other anchors as its negative examples. Denoting an observation indexed from dataset $\mathcal{O}$ as $o_i$, and its positive as $o_{i+}$, the logits can be written as $l_{i,j+} = p_i W \bar{c}_{j+}$; our loss function in practice is:

$$\mathcal{L}^{ATC} = -\mathbb{E}_{\mathcal{O}} \left[ \log \frac{\exp l_{i,i+}}{\sum_{o_j \in \mathcal{O}} \exp l_{i,j+}} \right]. \tag{2}$$

## 4 EXPERIMENTS

### 4.1 EVALUATION ENVIRONMENTS AND ALGORITHMS

We evaluate ATC on three standard, visually diverse RL benchmarks - the DeepMind control suite (DMControl; Tassa et al. 2018), Atari games in the Arcade Learning Environment (Bellemare et al., 2013), and DeepMind Lab (DMLab; Beattie et al. 2016). Atari requires discrete control in arcade-style games. DMControl is comprised of continuous control robotic locomotion and manipulation tasks. In contrast, DMLab requries the RL agent to reason in more visually complex 3D maze environments with partial observability.

We use ATC to enhance both on-policy and off-policy RL algorithms. For DMControl, we use RAD-SAC (Laskin et al., 2020a; Haarnoja et al., 2018) with the augmentation of Kostrikov et al. (2020), which randomly shifts the image in each coordinate (by up to 4 pixels), replicating edge pixel values as necessary to restore the original image size. A difference from prior work is that we use more downsampling in our convolutional network, by using strides $(2, 2, 2, 1)$ instead of $(2, 1, 1, 1)$ to reduce the convolution output image by 25x.[1] For both Atari and DMLab, we use PPO (Schulman et al., 2017). In Atari, we use feed-forward agents, sticky actions, and no end-of-life boundaries for RL episodes. In DMLab we used recurrent, LSTM agents receiving only a single time-step image input, the four-layer convolution encoder from Jaderberg et al. (2019), and we tuned the entropy bonus for each level. In the online setting, the ATC loss is trained using small replay buffer of recent experiences.

We include all our own baselines for fair comparison and provide complete settings in an appendix. Unless otherwise noted, each curve represents a minimum of 3 random seeds. The bold lines show the average, and the lightly shaded area around each curve represents the maximum extent of the best and worst seeds at each checkpoint.

### 4.2 ONLINE RL WITH ATC

**DMControl**   In the online setting, we found ATC to be capable of training the encoder by itself (*i.e.*, with encoder fully detached from any RL gradient update), achieving essentially equal or better scores versus end-to-end RL in all six environments we tested, Figure 2. In CARTPOLE-SWINGUP-SPARSE, where rewards are only received once the pole reaches vertical, ATC training enabled the agent to master the task significantly faster. The encoder is trained with one update for every RL update to the policy, using the same batch size, except in CHEETAH-RUN, which required twice the ATC updates.

**DMLab**   We experimented with two kinds of levels in DMLab: EXPLORE_GOAL_LOCATIONS, which requires repeatedly navigating a maze whose layout is randomized every episode, and LASERTAG_THREE_OPPONENTS, which requires fast reflexes to pursue and tag enemies at a distance. We found ATC capable of training fully detached encoders while achieving equal or better performance than end-to-end RL. Results are shown in Figure 3. Both environments exhibit spar-

---

[1]For our input image size $84 \times 84$, the convolution output image is $7 \times 7$ rather than $35 \times 35$. Performance remains largely unchanged, except for a small decrease in the HALF-CHEETAH environment, but the experiments run significantly faster and use less GPU memory.

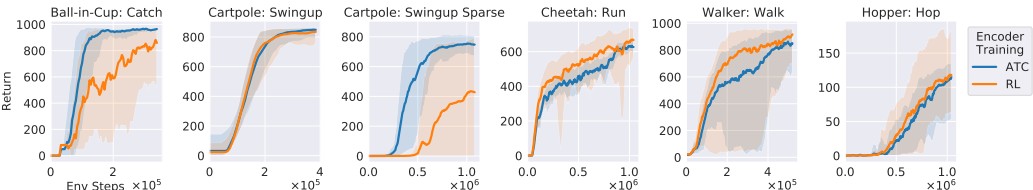

Figure 2: Online encoder training by ATC, fully detached from RL training, performs as well as end-to-end RL in DMControl, and better in sparse-reward environments (environment steps shown, see appendix for action repeats). Each curve is 10 random seeds.

sity which is greater in the "large" version than the "small" version, which our algorithm addresses, discussed next.

In EXPLORE, the goal object is rarely seen, especially early on, making its appearance difficult to learn. We therefore introduced prioritized sampling for ATC , with priorities corresponding to empirical absolute returns: $p \propto 1 + R_{abs}$, where $R_{abs} = \sum_{t=0}^{n} \gamma^t |r_t|$, to train more frequently on more informative scenes.[2] Whereas uniform-ATC performs slightly below RL, uniform-ATC outperforms RL and nearly matches using ATC (uniform) as an auxiliary task. By considering the encoder as a stand-alone feature extractor separate from the policy, no importance sampling correction is required.

In LASERTAG, enemies are often seen, but the reward of tagging one is rarely achieved by the random agent. ATC learns the relevant features anyway, boosting performance while the RL-only agent remains at zero average score. We found that increasing the rate of UL training to do twice as many updates[3] further improved the score to match the ATC-auxiliary agent, showing flexibility to address the representation-learning bottleneck when opponents are dispersed.

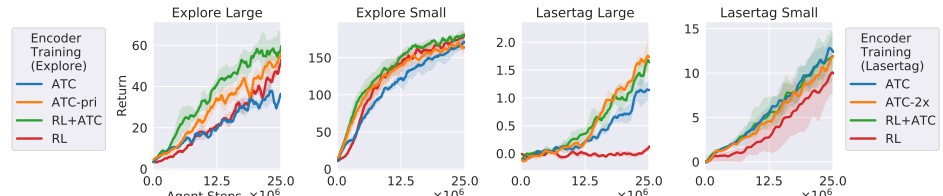

Figure 3: Online encoder training by ATC, fully detached from the RL agent, performs as well or better than end-to-end RL in DMLab (1 agent step = 4 environment steps, the standard action repeat). Prioritized ATC replay (EXPLORE) or increased ATC training (LASERTAG) addresses sparsities to nearly match performance of RL with ATC as an auxiliary loss (RL+ATC). Each curve is 3 random seeds.

**Atari** We tested a diverse subset of eight Atari games, shown in Figure 4. We found detached-encoder training to work as well as end-to-end RL in five games, but performance suffered in BREAKOUT and SPACE INVADERS in particular. Using ATC as an auxiliary task, however, improves performance in these games and others. We found it helpful to anneal the amount of UL training over the course of RL in Atari (details in an appendix). Notably, we found several games, including SPACE INVADERS, to benefit from using ATC only to initialize encoder weights, done using an initial 100k transitions gathered with a uniform random policy. Some of our remaining experiments provide more insights into the challenges of this domain.

### 4.3 ENCODER PRE-TRAINING BENCHMARKS

To benchmark the effectiveness of different UL algorithms for RL, we propose a new evaluation methodology that is similar to how UL pre-training techniques are measured in computer vision (see

---

[2]In EXPLORE_GOAL_LOCATIONS, the only reward is +10, earned when reaching the goal object.
[3]Since the ATC batch size was 512 but the RL batch size was 1024, performing twice as many UL updates still only consumed the same amount of encoder training data as RL. We did not fine-tune for batch size.

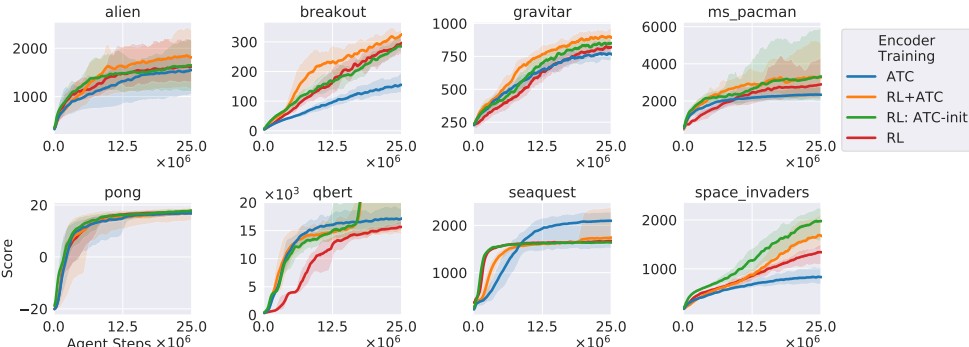

Figure 4: Online encoder training using ATC, fully detached from the RL agent, works well in 5 of 8 games tested (1 agent step = 4 environment steps, the standard action repeat). 6 of 8 games benefit significantly from using ATC as an auxiliary loss or for weight initialization. Each curve is 8 random seeds.

*e.g.* Chen et al. (2020); Grill et al. (2020)): (i) collect a data set composed of expert demonstrations from each environment; (ii) pre-train the CNN encoder with that data offline using UL; (iii) evaluate by using RL to learn a control policy while keeping the encoder weights frozen. This procedure isolates the asymptotic performance of each UL algorithm for RL. For convenience, we drew expert demonstrations from partially-trained RL agents, and every UL algorithm trained on the same data set for each environment. Our RL agents used the same post-encoder architectures as in the online experiments. Further details about pre-training by each algorithm are provided in an appendix.

**DMControl** We compare ATC against two competing algorithms: Augmented Contrast (AC), from CURL (Laskin et al., 2020b), which uses the same observation for the anchor and the positive, and a VAE (Kingma & Welling, 2013), for which we found better performance by introducing a time delay to the target observation (VAE-T). We found ATC to match or outperform the other algorithms, in all four test environments, as shown in Figure 5. Further, ATC is the only one to match or outperform the reference end-to-end RL across all cases.

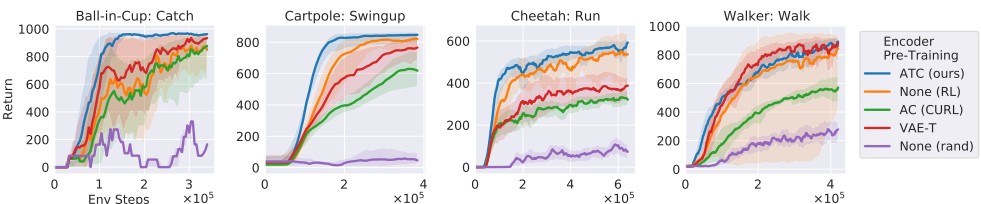

Figure 5: RL in DMControl, using encoders pre-trained on expert demonstrations using UL, with weights frozen—across all domains, ATC outperforms prior methods and the end-to-end RL reference. Each curve is a mininum of 4 random seeds.

**DMLab** We compare against both Pixel Control (Jaderberg et al., 2017; Hessel et al., 2019) and CPC (van den Oord et al., 2018), which have been shown to bring strong benefits in DMLab. While all algorithms perform similarly well in EXPLORE, ATC performs significantly better in LASERTAG, Figure 6. Our algorithm is simpler than Pixel Control and CPC in the sense that it uses neither actions, deconvolution, nor recurrence.

**Atari** We compare against Pixel Control, VAE-T, and a basic inverse model which predicts actions between pairs of observations. We also compare

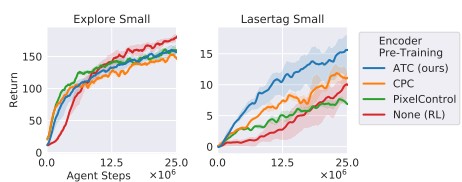

Figure 6: RL in DMLab, using pre-trained encoders with weights frozen–in LASERTAG especially, ATC outperforms leading prior UL algorithms.

against Spatio-Temporal Deep InfoMax (ST-DIM), which uses temporal contrastive losses with "local" features from an intermediate convolution layer to ensure attention to the whole screen; it was shown to produce detailed game-state knowledge when applied to individual frames (Anand et al., 2019). Of the four games shown in Figure 7, ATC is the only UL algorithm to match the end-to-end RL reference in GRAVITAR and BREAKOUT, and it performs best in SPACE INVADERS.

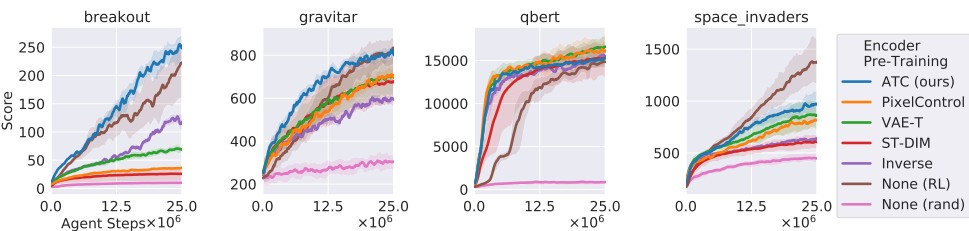

Figure 7: RL in Atari, using pre-trained encoders with weights frozen—ATC outperforms several leading, prior UL algorithms and exceeds the end-to-end RL reference in 3 of the 4 games tested.

## 4.4 MULTI-TASK ENCODERS

In the offline setting, we conducted initial explorations into the capability of ATC to learn multi-task encoders, simply by pre-training on demonstrations from multiple environments. We evaluate the encoder by using it, with frozen weights, in separate RL agents learning each downstream task.

**DMControl** Figure 8 shows our results in DMControl, where we pretrained using only the four environments in the top row. Although the encoder was never trained on the HOPPER, PENDULUM, nor FINGER domains, the multi-task encoder supports efficient RL in them. PENDULUM-SWINGUP and CARTPOLE-SWINGUP-SPARSE stand out as challenging environments which benefited from cross-domain and cross-task pre-training, respectively. The pretraining was remarkably efficient, requiring only 20,000 updates to the encoder.

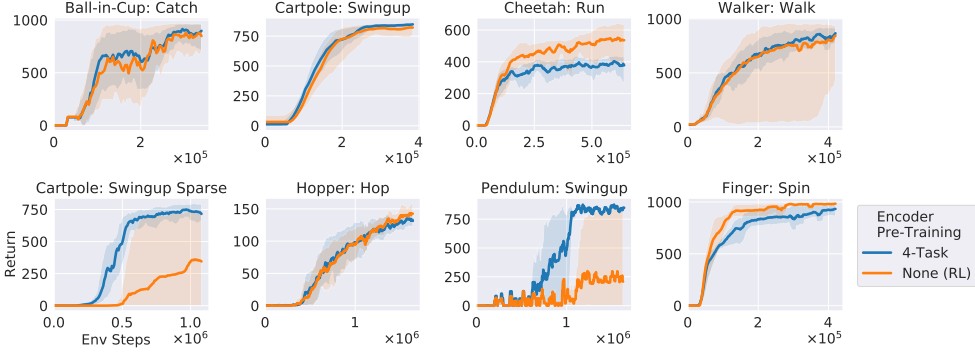

Figure 8: Separate RL agents using a single encoder with weights frozen after pre-training on expert demonstrations from the four top environments. The encoder generalizes to four new environments, bottom row, where sparse reward tasks especially benefit from the transfer. Each curve is minimum 4 random seeds.

**Atari** Atari proved a more challenging domain for learning multi-task encoders. Learning all eight games together in Figure 11, in the appendix, resulted in diminished performance relative to single-game pretraining in three of the eight. The decrease was partially alleviated by widening the encoder with twice as many filters per layer, indicating that representation capacity is a limiting factor. To test generalization, we conducted a seven-game pre-training experiment where we test the encoder on the held-out game. Most games suffered diminished performance (although still perform significantly higher than a frozen random encoder), confirming the limited extent to which visual features transfer across these games.

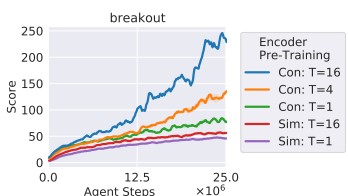

Figure 9: BREAKOUT benefits from contrasting against negatives from several neighboring time steps.

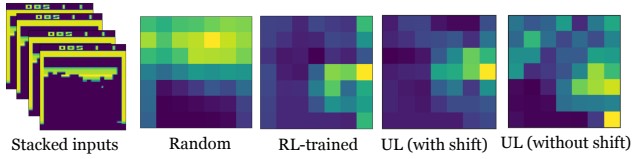

Figure 10: An example scene from BREAKOUT, where a low-performance UL encoder (without shift) focuses on the paddle. Introducing random shift and sequence data makes the high-performance UL encoder (full ATC) focus near the ball, as does the encoder from a fully-trained, end-to-end RL agent.

### 4.5    ABLATIONS AND ENCODER ANALYSIS

**Random Shift in ATC**    In offline experiments, we discovered random shift augmentations to be helpful in all domains. To our knowledge, this is the first application of random shift to 3D visual environments as in DMLab. In Atari, we found performance in GRAVITAR to suffer from random shift, but reducing the probability of applying random shift to each observation from 1.0 to 0.1 alleviated the effect while still bringing benefits in other games, so we used this setting in our main experiments. Results are shown in Figure 12 in an appendix.

**Random Shift in RL**    In DMControl, we found the best results when using random shift during RL, even when training with a frozen encoder. This is evidence that the augmentation regularizes not only the representation but also the policy, which first processes the latent image into a 50-dimensional vector. To unlock computation and memory benefits of replaying only the latent images for the RL agent, we attempted to apply data augmentation to the latent image. But we found the smallest possible random shifts to be too extreme. Instead, we introduce a new augmentation, *subpixel random shift*, which linearly interpolates among neighboring pixels. As shown in Figure 13 in the appendix, this augmentation restores performance when applied to the latent images, allowing a pre-trained encoder to be entirely bypassed during policy training updates.

**Temporal Contrast on Sequences**    In BREAKOUT alone, we discovered that composing the UL training batch of trajectory segments, rather than individual transitions, gave a significant benefit. Treating all elements of the training batch independently provides "hard" negatives, since the encoder must distinguish between neighboring time steps. This setting had no effect in the other Atari games tested, and we found equal or better performance using individual transitions in DMControl and DMLab. Figure 9 further shows that using a similarity loss (Grill et al., 2020) does not capture the benefit.

**Encoder Analysis**    We analyzed the learned encoders in BREAKOUT to further study this ablation effect. Similar to Zagoruyko & Komodakis (2016), we compute spatial attention maps by mean-pooling the absolute values of the activations along the channel dimension and follow with a 2-dimensional spatial softmax. Figure 10 shows the attention of four different encoders on the displayed scene. The poorly performing UL encoder heavily utilizes the paddle to distinguish the observation. The UL encoder trained with random shift and sequence data, however, focuses near the ball, as does the fully-trained RL encoder. (The random encoder mostly highlights the bricks, which are less relevant for control.) In an appendix, we include other example encoder analyses from Atari and DMLab which show ATC-trained encoders attending only to key objects on the game screen, while RL-trained encoders additionally attend to potentially distracting features such as game score.

## 5    CONCLUSION

Reward-free representation learning from images provides flexibility and insights for improving deep RL agents. We have shown a broad range of cases where our new unsupervised learning algorithm can fully replace RL for training convolutional encoders while maintaining or improving

online performance. In a small number of environments–a few of the Atari games–including the RL loss for encoder training still surpasses our UL-only method, leaving opportunities for further improvements in UL for RL.

Our preliminary efforts to use actions as inputs (into the *predictor* MLP) or as prediction outputs (inverse loss) with ATC did not immediately yield strong improvements. We experimented only with random shift, but other augmentations may be useful, as well. In multi-task encoder training, our technique avoids any need for sophisticated reward-balancing (Hessel et al., 2019), but more advanced training methods may still help when the required features are in conflict, as in Atari, or if they otherwise impact our loss function unequally. On the theoretical side, it may be helpful to analyze the effects of domain shift on the policy when a detached representation is learned online.

One obvious application of our offline methodology would be in the batch RL setting, where the agent learns from a fixed data set. Our offline experiments showed that a relatively small number of transitions are sufficient to learn rich representations by UL, and the lower limit could be further explored. Overall, we hope that our algorithm and experiments spur further developments leveraging unsupervised learning for reinforcement learning.

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

# A  APPENDIX

## A.1  ALGORITHMS

---

**Algorithm 1**
Online RL with decoupled ATC encoder (steps distinct from end-to-end RL in blue)

---

**Require:** $\theta_{ATC}, \phi_\pi$      ▷ ATC model parameters (encoder $f_\theta$ thru contrast $W$), policy parameters
  1:   $\mathcal{S} \leftarrow \{\}$                                                      ▷ replay buffer of observations
  2:   $\bar{\theta}_{ATC} \leftarrow \theta_{ATC}$                            ▷ initialize momentum encoder (conv and linear only)
  3: **repeat**
  4:      Sample environment and policy, through encoder:
  5:      **for** 1 to m **do**                                         ▷ a minibatch
  6:         $a \sim \pi(\cdot|f_\theta(s); \phi), s' \sim T(s,a), r \sim R(s,a,s')$
  7:         $\mathcal{S} \leftarrow \mathcal{S} \cup \{s\}$                 ▷ store observations (delete oldest if full)
  8:         $s \leftarrow s'$
  9:      **end for**
10:      Update policy by given RL formula:                  ▷ on- or off-policy
11:      **for** 1 to n **do**              ▷ given number RL updates per minibatch
12:         $\phi_\pi \leftarrow \phi_\pi + RL(s,a,s',r; \phi_\pi)$          ▷ stop gradient into encoder
13:      **end for**
14:      Update encoder (and contrastive model) by ATC:
15:      **for** 1 to p **do**
16:         $s, s_+ \sim \mathcal{S}$                 ▷ sample observations: anchors and positives
17:         $\theta_{ATC} \leftarrow \theta_{ATC} - \lambda_{ATC} \nabla_{\theta_{ATC}} \mathcal{L}^{ATC}(s, s_+)$     ▷ ATC gradient update
18:         $\bar{\theta}_{ATC} \leftarrow (1-\tau)\bar{\theta}_{ATC} + \tau\theta_{ATC}$    ▷ update momentum encoder (conv and linear only)
19:      **end for**
20: **until** converged
21: **return** Encoder $f_\theta$ and policy $\pi_\phi$

---

## A.2  ADDITIONAL FIGURES

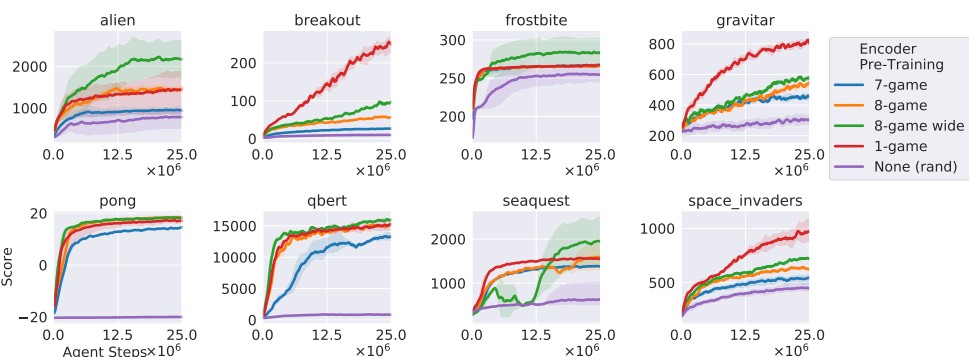

Figure 11: RL using multi-task encoders (all with weights frozen) for eight Atari games gives mixed performance, partially improved by increased network capacity (8-game-wide). Training on 7 games and testing on the held-out one yields diminished but non-zero performance, showing some limited feature transfer between games.

In subpixel random shift, new pixels are a linearly weighted average of the four nearest pixels to a randomly chosen coordinate location. We used uniformly random horizontal and vertical shifts, and tested maximum displacements in $(\pm) \{0.1, 0.25, 0.5, 0.75, 1.0\}$ pixels (with "edge" mode padding $\pm 1$). We found $0.5$ to work well in all tested domains, restoring the performance of raw image augmentation but eliminating convolutions entirely from the RL training updates.

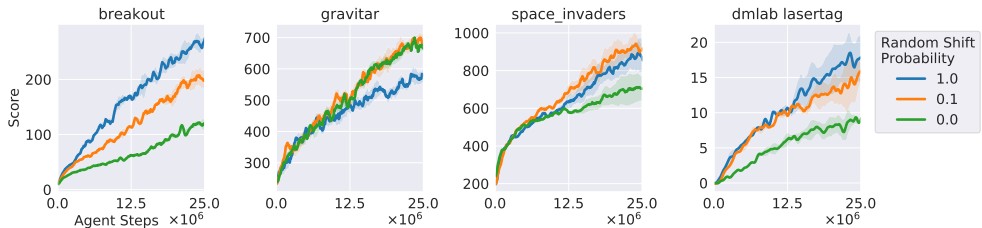

Figure 12: Random shift augmentation helps in some Atari games and hurts in others, but applying with probability 0.1 is a performant middle ground. DMLab benefits from random shift. (Offline pre-training.)

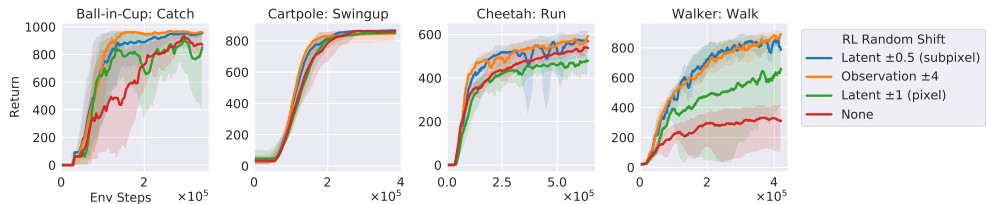

Figure 13: Even after pre-training encoders for DMControl using random shift, RL requires augmentation—our subpixel augmentation acts on the (compressed) latent image, permitting its use in the replay buffer.

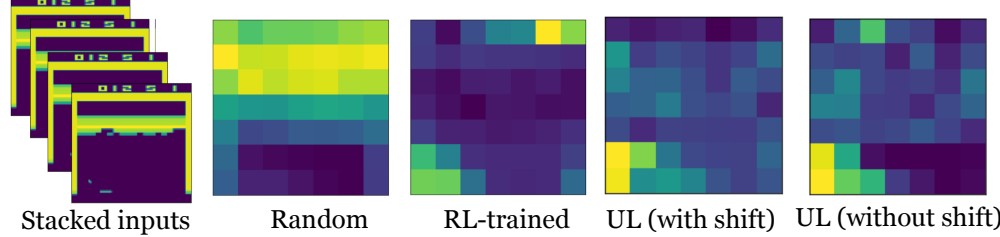

Figure 14: Attention map in BREAKOUT which shows the RL-trained encoder focusing on game score, whereas UL ATC encoder focuses properly on the paddle and ball.

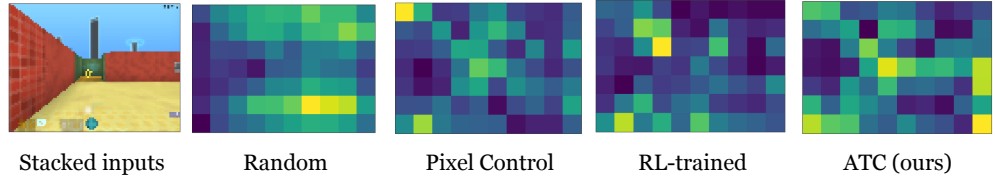

Figure 15: Attention map in LASERTAG. UL encoder with pixel control focuses on the score, while UL encoder with the proposed ATC focuses properly on the coin similar to RL-trained encoder.

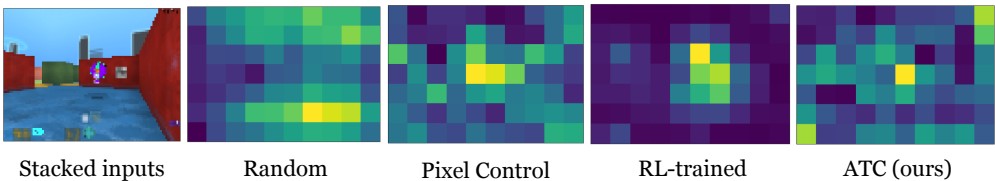

Figure 16: Attention map in the LASERTAG which shows that UL encoders focus properly on the enemy similar to RL-trained encoder.

## A.3 RL SETTINGS

Table 1: DMControl, RAD-SAC Hyperparameters.

| HYPERPARAMETER | VALUE |
|---|---|
| OBSERVATION RENDERING | $(84, 84)$, RGB |
| RANDOM SHIFT PAD | $\pm 4$ |
| REPLAY BUFFER SIZE | 1e5 |
| INITIAL STEPS | 1e4 |
| STACKED FRAMES | 3 |
| ACTION REPEAT | 2 (FINGER, WALKER) |
| | 8 (CARTPOLE) |
| | 4 (REST) |
| OPTIMIZER | ADAM |
| $(\beta_1, \beta_2) \rightarrow (f_\theta, \pi_\psi, Q_\phi)$ | $(.9, .999)$ |
| $(\beta_1, \beta_2) \rightarrow (\alpha)$ | $(.5, .999)$ |
| LEARNING RATE $(f_\theta, \pi_\psi, Q_\phi)$ | 2e−4 (CHEETAH) |
| | 1e−3 (REST) |
| LEARNING RATE $(\alpha)$ | 1e−4 |
| BATCH SIZE | 512 (CHEETAH, PENDULUM) |
| | 256 (REST) |
| $Q$ FUNCTION EMA $\tau$ | 0.01 |
| CRITIC TARGET UPDATE FREQ | 2 |
| CONVOLUTION FILTERS | $[32, 32, 32, 32]$ |
| CONVOLUTION STRIDES | $[2, 2, 2, 1]$ |
| CONVOLUTION FILTER SIZE | 3 |
| ENCODER EMA $\tau$ | 0.05 |
| LATENT DIMENSION | 50 |
| HIDDEN UNITS (MLP) | $[1024, 1024]$ |
| DISCOUNT $\gamma$ | .99 |
| INITIAL TEMPERATURE | 0.1 |

Table 2: Atari, PPO Hyperparameters.

| HYPERPARAMETER | VALUE |
|---|---|
| OBSERVATION RENDERING | $(84, 84)$, GREY |
| STACKED FRAMES | 4 |
| ACTION REPEAT | 4 |
| OPTIMIZER | ADAM |
| LEARNING RATE | 2.5e−4 |
| PARALLEL ENVIRONMENTS | 16 |
| SAMPLING INTERVAL | 128 |
| LIKELIHOOD RATIO CLIP, $\epsilon$ | 0.1 |
| PPO EPOCHS | 4 |
| PPO MINIBATCHES | 4 |
| CONVOLUTION FILTERS | $[32, 64, 64]$ |
| CONVOLUTION FILTER SIZES | $[8, 4, 3]$ |
| CONVOLUTION STRIDES | $[4, 2, 1]$ |
| HIDDEN UNITS (MLP) | $[512]$ |
| DISCOUNT $\gamma$ | .99 |
| GENERALIZED ADVANTAGE ESTIMATION $\lambda$ | 0.95 |
| LEARNING RATE ANNEALING | LINEAR |
| ENTROPY BONUS COEFFICIENT | 0.01 |
| EPISODIC LIVES | FALSE |
| REPEAT ACTION PROBABILITY | 0.25 |
| REWARD CLIPPING | $\pm 1$ |
| VALUE LOSS COEFFICIENT | 1.0 |

Table 3: DMLab, PPO Hyperparameters.

| HYPERPARAMETER | VALUE |
|---|---|
| OBSERVATION RENDERING | $(72, 96)$, RGB |
| STACKED FRAMES | 1 |
| ACTION REPEAT | 4 |
| OPTIMIZER | ADAM |
| LEARNING RATE | 2.5e$-$4 |
| PARALLEL ENVIRONMENTS | 16 |
| SAMPLING INTERVAL | 128 |
| LIKELIHOOD RATIO CLIP, $\epsilon$ | 0.1 |
| PPO EPOCHS | 1 |
| PPO MINIBATCHES | 2 |
| CONVOLUTION FILTERS | $[32, 64, 64, 64]$ |
| CONVOLUTION FILTER SIZES | $[8, 4, 3, 3]$ |
| CONVOLUTION STRIDES | $[4, 2, 1, 1]$ |
| HIDDEN UNITS (LSTM) | $[256]$ |
| SKIP CONNECTIONS | CONV 3, 4; LSTM |
| DISCOUNT $\gamma$ | .99 |
| GENERALIZED ADVANTAGE ESTIMATION $\lambda$ | 0.97 |
| LEARNING RATE ANNEALING | NONE |
| ENTROPY BONUS COEFFICIENT | 0.01 (EXPLORE) |
| | 0.0003 (LASERTAG) |
| VALUE LOSS COEFFICIENT | 0.5 |

## A.4 ONLINE ATC SETTINGS

Table 4: Common ATC Hyperparameters.

| HYPERPARAMETER | VALUE |
| --- | --- |
| RANDOM SHIFT PAD | $\pm 4$ |
| LEARNING RATE | 1e−3 |
| LEARNING RATE ANNEALING | COSINE |
| TARGET UPDATE INTERVAL | 1 |
| TARGET UPDATE $\tau$ | 0.01 |
| PREDICTOR HIDDEN SIZES, $h_\psi$ | [512] |
| REPLAY BUFFER SIZE | 1e5 |

Table 5: DMControl ATC Hyperparameters.

| HYPERPARAMETER | VALUE |
| --- | --- |
| RANDOM SHIFT PROBABILITY | 1 |
| BATCH SIZE | AS RL (INDIVIDUAL OBSERVATIONS) |
| TEMPORAL SHIFT, $k$ | 1 |
| MIN AGENT STEPS TO UL | 1e4 |
| MIN AGENT STEPS TO RL | 1e4 |
| UL UPDATE SCHEDULE | AS RL |
| | (2X CHEETAH) |
| LATENT SIZE | 128 |

Table 6: Atari ATC Hyperparameters.

| HYPERPARAMETER | VALUE |
| --- | --- |
| RANDOM SHIFT PROBABILITY | 0.1 |
| BATCH SIZE | 512 (32 TRAJECTORIES OF 16 TIME STEPS) |
| TEMPORAL SHIFT, $k$ | 3 |
| MIN AGENT STEPS TO UL | 5e4 |
| MIN AGENT STEPS TO RL | 1e5 |
| UL UPDATE SCHEDULE | ANNEALED QUADRATICALLY FROM 6 PER SAMPLER ITERATION |
| | (1e4 ONCE AT 1e5 STEPS FOR WEIGHT INITIALIZATION) |
| LATENT SIZE | 256 |

Table 7: DMLab ATC Hyperparameters.

| HYPERPARAMETER | VALUE |
| --- | --- |
| RANDOM SHIFT PROBABILITY | 1 |
| BATCH SIZE | 512 (INDIVIDUAL OBSERVATIONS) |
| TEMPORAL SHIFT, $k$ | 3 |
| MIN AGENT STEPS TO UL | 5e4 |
| MIN AGENT STEPS TO RL | 1e5 |
| UL UPDATE SCHEDULE | 2 PER SAMPLER ITERATION |
| LATENT SIZE | 256 |

### A.5 Offline Pre-Training Details

We conducted coarse hyperparameter sweeps to tune each competing UL algorithm. In all cases, the best setting is the one shown in our comparisons.

When our VAEs include a time difference between input and reconstruction observations, we include one hidden layer with action additionally input between the encoder and decoder. We tried both 1.0 and 0.1 KL-divergence weight in the VAE loss, and found 0.1 to perform better in both DMControl and Atari.

**DMControl** For the VAE, we experimented with 0 and 1 time step difference between input and reconstruction target observations and training for either 1e4 or 5e4 updates. The best settings were 1-step temporal, and 5e4 updates, with batch size 128. ATC used 1-step temporal, 5e4 updates (although this can be significantly decreased), and batch size 256 (including CHEETAH). The pretraining data set consisted of the first 5e4 transitions from a RAD-SAC agent learning each task, including 5e3 random actions. Within this span, CARTPOLE and BALL_IN_CUP learned completely, but WALKER and CHEETAH reached average returns of 514 and 630, respectively (collected without the compressive convolution).

**DMLab** For Pixel Control, we used the settings from Hessel et al. (2019) (see the appendix therein), except we used only empirical returns, computed offline (without bootstrapping). For CPC, we tried training batch shapes, $batch \times time$ in (64, 8), (32, 16), (16, 32), and found the setting with rollouts of length 16 to be best. We contrasted all elements of the batch against each other, rather than only forward constrasts. In all cases we also used 16 steps to warmup the LSTM. For all algorithms we tried learning rates $3e-4$ and $1e-3$ and both 5e4 and 1.5e5 updates. For ATC and CPC, the lower learning rate and higher number of updates helped in LASERTAG especially. The pretraining data was 125e3 samples from partially trained RL agents receiving average returns of 127 and 6 in EXPLORE_GOAL_LOCATIONS_SMALL and LASERTAG_THREE_OPPONENTS_SMALL, respectively.

**Atari** For the VAE, we experimented with 0, 1, and 3 time step difference between input and reconstruction target, and found 3 to work best. For ST-DIM we experimented with 1, 3, and 4 time steps differences, and batch sizes from 64 to 256, learning rates $1e-3$ and $5e-4$. Likewise, 3-step delay worked best. For the inverse model, we tried 1- and 3-step predictions, with 1-step working better overall, and found random shift augmentation to help. For pixel control, we used the settings in Jaderberg et al. (2017), again with full empirical returns. We ran each algorithm for up to 1e5 updates, although final ATC results used 5e4 updates. We ran each RL agent with and without observation normalization on the latent image and observed no difference in performance. Pretraining data was 125e3 samples sourced from the replay buffer of DQN agents trained for 15e6 steps with epsilon-greedy $\epsilon = 0.1$. Evaluation scores were:

Table 8: Atari Pre-Training Data Source Agents.

| GAME | EVALUATION SCORE |
|---:|:---|
| ALIEN | $1,800$ |
| BREAKOUT | 279 |
| FROSTBITE | $1,400$ |
| GRAVITAR | 390 |
| PONG | 18 |
| QBERT | $8,800$ |
| SEAQUEST | $11,000$ |
| SPACE INVADERS | $1,200$ |

