# OpenReview forum: "Decoupling Representation Learning from Reinforcement Learning"
_ICLR.cc/2021/Conference — Reject_

### Official Review · AnonReviewer4 · 2020-10-17
**A good improvement over prior unsupervised RL tasks in many setups**

**Rating:** 7
**Confidence:** 3

**Review:**

The paper introduces an unsupervised task called Augmented Temporal Contrast which associates pairs of observations separated in time using a contrastive loss. The paper uses the task in several training regimes (in online RL, pretraining and multi-task RL).

Pros:
- Well written and structured paper.
- Interesting, general and simple to implement task.
- Evaluated in several training regimes and on several environments.
- Improves sample efficiency on most of the environments and setups, and improves over prior methods.
- The attention maps in the paper and appendix are great and although they may be hand picked(?) examples, it highlights the issue with many approaches that can't model a goal rarely seen.

Cons:
- You write you use multiple seeds but I don't see anywhere details on this? Consider adding it to the tables in appendix.
- In Figure 3 one agent step is 4x environment frames? I suggest to make it clear in plot or in caption. In Figure 2, environment frames is used.

Comments/questions:
- Wrt. 4.1, to what extent is the "small" replay for DMLab necessary over just using observations in batches/unrolls? Looking at Table 3 I can't see how large the replay is? 10k as in SAC or smaller?

Update: Not all the experiments are particular thorough and the novelty less than expected.

---

> ### Author Response · Authors · 2020-11-19
> **Appreciate the positive assessment!**
>
>  We appreciate your positive assessment of our paper. As you and other reviewers (R2 and R3) mentioned, we introduce a novel unsupervised learning method for visual RL with extensive experiments on various RL benchmarks. We hope that our work will bring novel ideas/perspectives and highly effective techniques towards improving visual RL to the ICLR community.
>
> Please see the following clarifications in blue text in the new draft:
>
> ---
> **Q1. Random seeds**
>
> A1. Good point!  We ran at least 3 for each curve, and have added this to the paper.
>
> ---
> **Q2. Frame repeat**
>
> A2. Thanks, we have clarified this relationship in the captions in the draft (the conventions in previous literature in DMControl is to report environment steps, and Atari/PPO more often agent steps).
>
> ---
> **Q3. Replay Buffer size**
>
> A3. Thanks we have now listed this in the appendix for Online ATC settings.  We used 100k in all environments (as standard in SAC, and much smaller than the 1M standard in DQN).  This would be an interesting ablation to decrease the size or try to do away with it altogether for PPO.  DMLab slightly preferred not to have time-correlated anchors in each training batch but it wasn’t a huge effect, and Atari was fine with it.  But keeping a replay buffer does provide more flexibility for techniques like prioritized replay or simply doing more representation learning updates, both of which were helpful in DMLab.

---

### Official Review · AnonReviewer3 · 2020-10-26
**A new unsupervised learning method for learning latent representations for visual control domains.**

**Rating:** 5
**Confidence:** 3

**Review:**

**Summary:**
This paper presents a new unsupervised learning method for learning latent representations for visual RL control domains. The method, Augmented Temporal Contrast (ATC), can be used alone to learn a representation to be combined with an RL algorithm, or as an auxiliary task in an end-to-end system. ATC matches or outperforms comparable end-to-end systems in several environments. The paper provides an extensive experimental study to support its claims.

**Strengths:**
The paper is clearly written, and all of the main points are well articulated. ATC appears sufficiently novel, and is applicable to a wide variety of domains, and can be deployed in various configurations (e.g. auxiliary task, unsupervised pre-training, etc…). Included is a thorough experimental study that effectively demonstrates the performance of the method.

**Weaknesses:**
Although the experimental study seem thorough. I could not find the actual number of independent runs (seeds) for each domain listed anywhere. This information should be included so that the reader can better evaluate the variance of each method, and make more confident conclusions.

**Recommendation:**
Overall I vote to accept. The method presented in the paper is not revolutionary, but it appears to be novel and significant enough to be of interest to deep RL practitioners.

**Questions:**
How many independent runs (seeds) were used in each of the domains? Can this information be included in the main text?

**After Author Response and Discussion:**
Thanks to the authors for their responses. After reading the other reviews and the author responses, I am lowering my score to 5. I think that the number of independent runs used (especially on the smaller domains), and the way the results are presented with the min-max extent makes me less convinced of the results than I was in the initial review. Adding many more independent runs (seeds), especially on the smaller domains, would improve my confidence a lot. Overall I think the paper is of interest to the community, but the experiments and their analysis could be improved.

---

> ### Author Response · Authors · 2020-11-19
> **Appreciate the positive assessment!**
>
> We appreciate your positive assessment of our paper. As you and other reviewers (R2 and R4) mentioned, we introduce a novel unsupervised learning method for visual RL with extensive experiments on various RL benchmarks. We hope that our work will bring novel ideas/perspectives and highly effective techniques towards improving visual RL to the ICLR community.
>
> ---
> **Q. Random seeds**
>
> A.  Good point!  We ran at least 3 for each curve, and have added this to the paper. (blue text in the updated draft)

---

> > ### Comment · AnonReviewer3 · 2020-11-19
> > **3 seeds is probably too few to draw reliable and confident conclusions**
> >
> > When I read the paper initially, I assumed that there were more independent runs (seeds) for each experiment than just 3. This fact makes me much less confident about the significance of the experimental results. Observing the same trends in the results with at least 10 seeds would greatly increase my confidence in the conclusions.
> >
> > Further I think the way that the learning curves are presented in the plots with the minimum and maximum extent could be misleading. With only 3 curves, the most transparent way to present the data would be to simple plot all of them on the same plot (maybe with the mean). This gives a good picture of the variance in each run, and would highlight any modes that are apparent during learning. Do some runs learn fast then plateau early? Do some runs learn slow, but do better eventually? Do some runs flatline? All this interesting information can be hidden if the data is summarised incorrectly. This presentation style works well with more runs (10+) also.
> >
> > Would it be possible to increase the number of seeds reported in the paper? If you can't add more seeds, do you think just plotting all the learning curves, without the minimum and maximum extents, would be a valid way to present your data?

---

> > > ### Author Response · Authors · 2020-11-21
> > > **yes, will provide more seeds**
> > >
> > > Yes, good point to have more seeds.  We are launching more experiments to improve the confidence levels.
> > >
> > > In the meantime, here is a more full accounting of the number of seeds already in the paper; hopefully it comes across as a large total number of random seeds given the many env/algo combinations (apologies for the confusion that we conservatively reported as 3 seeds some experiments which were actually more, notably DMControl online, our mistake!).  We'll  also note that we ran many experiments leading up to the final ones in the paper, which gives us confidence about the results (of course doesn't count in the paper).
> > >
> > >
> > > In order in the paper:
> > > DMControl online -- 6 seeds
> > > Atari online -- 4 seeds
> > > DMLab online -- 3 seeds (except lasertag_small -- 5 seeds)
> > >
> > > DMcontrol offline -- 4 seeds
> > > Atari offline -- minimum 3 seeds
> > > DMLab offline -- ATC is 4 seeds, PC and CPC are 2 seeds each
> > >
> > > DMControl multi-env -- 4 seeds ATC (6 seeds RL)
> > > Atari multi-env -- 3 seeds (except 8-game pretraining -- 2 seeds)
> > >
> > >
> > > We will give it a try to plot each individual run, which would be the best raw presentation!  Just afraid this will get cluttered with the DMLab and Atari experiments which have so many different algos on the same plot.

---

> > > ### Author Response · Authors · 2020-11-25
> > > **more seeds run**
> > >
> > > We've increased the number of seeds for online RL vs ATC experiments from 6 to 10 in DMControl and from 4 to 8 in Atari, without significant change in outcome.  (new plots are in the paper Fig 2 and Fig 4)  DMLab experiments are underway now, altho they take longer to run.

---

### Official Review · AnonReviewer1 · 2020-10-27
**A well-written paper with sound reasoning and mostly good results that should sharpen and narrow its focus**

**Rating:** 5
**Confidence:** 4

**Review:**

Summary of the paper:
The paper aims to define an unsupervised pretraining architecture that can be used to pretrain representations for a reinforcement learning agent. The proposed solution is called "Augmented Temporal Contrast" (ATC). It consists of an encoder-compressor-predictor architecture for an (augmented) observation that is trained in a small latent space by minimizing the InfoNCE loss between that prediction, and the encoding of some future state.
The representation used (the pretrained encoder) is then frozen, and a policy network is trained consuming these representations. The authors then evaluate their approach on several different tasks: a control domain (DMcontrol), Mazenagivation (DMLab) and Atari. They show that their approach - without finetuning the representations - achieves comparable results to end-to-end reinforcement learning on most of the different domains. They also show that adding their defined loss as a regularizer always helps learning. They finally show that their representations can generalize out-of-domain by running several multi-task experiments, while only having pretrained on one domain.

Commentary on the goal of the paper:

The goal of the paper is extremely important: separating learning representations from learning policies would enable better transfer, possible more sample efficiency and lower variance in outcome.

Strengths:
- The authors propose a well-designed solution that combines existing approaches in a well thought-out way.
- The paper is extremely well written
- The paper has an extensive empirical section.
- Results are generally very good.

Weaknesses:
- The goal of the paper is a bit vague. As I said above, I agree and understand the desire for the separation of representation learning and reinforcement learning. However, the paper would have been stronger if it had concentrated on a single benefit of this separation, and evaluated their approach on that. While they say that their goal is to investigate "how to learn representations which are agnostic to rewards", this is too general as well. (One example would be to say that the decoupling makes for better generalizes to new MDPs - but this is not the focus of the analysis, just an aspect. The paper is, unfortunately, less convincing for it.)
- Along a similar line of thought, the results, while strong, are not as convincing as they could be, because the paper does not focus on the benefits of reward-agnostic representations learned by ATC. The results that consider the generalization advantage (the Multi-Task learning experiments) are weaker, without the paper offering an analysis as to why.

In total, I would argue that this is a well-written paper with interesting analysis that could be a lot stronger by narrowing the scope of the contained argument. I argue for rejection, because I can see an updated version of this paper to be a great paper.

---

> ### Author Response · Authors · 2020-11-19
> **main goal -- show decoupled UL works as well as end-to-end RL //  will add more analysis in multi-domain case**
>
>
> We appreciate your thoughtful consideration of our paper. As other reviewers (R2, R3, R4) mentioned, we introduce a novel unsupervised learning method for visual RL with extensive experiments on various RL benchmarks. We hope that our work will bring novel ideas/perspectives and highly effective techniques towards improving visual RL to the ICLR community.
>
> ---
> **Q1. Goal of the paper**
>
> **A1.** The goal of the paper is to show that unsupervised representation learning can be decoupled from reinforcement learning without degrading the algorithm’s performance relative to end-to-end RL. To the best of our knowledge, we are the first to show that this is possible. We then investigate the properties of unsupervised pre-training through a series of detailed ablations. We note that not only do we present pre-training experiments, we first show ATC can successfully learn representations in an online fashion which starts from scratch with the RL agent.
>
> The questions of emphasis in the paper are interesting and worth discussing.  We would like to argue that the paper very specifically and intentionally addresses a capability that has been hinted at in a long series of related works, but never fully achieved.  For this reason it is a worthwhile contribution to explore a broad range of (standard) environments, for example to identify any corner cases which don’t quite work.  In the end, we discovered only a few of these (some of the Atari games).  Establishing and reporting all of this is already a substantial amount of new material and is the main point of the paper.  We can make this emphasis more explicit in the writing.
>
> ---
> **Q2. Analysis on generality of representation**
>
> **A2:** Agreed it would feel rather unsatisfying not to include at least some coverage of the immediate benefits of our approach outside of the usual RL workflow (i.e. not just ATC as an auxiliary task or detached encoder training).   We show that our method makes it quite easy to leverage multi-domain training in DMControl, but the same is not fully true for Atari.  This result clearly points to the fact that one environment can share visual features while the other cannot--a useful reference point for future research into multi-domain methods.
>
> **_Your suggestion to deepen the multi-domain studies could lead to a very interesting addition:_** to include encoder attention analysis for these cases (as we did for the ablations).  Thanks very much for pushing this direction, we will set about generating these in time for final review!
>
> ---
> **Q3. Several ideas in the paper**
>
> **A3.** Furthermore we were able to show that image augmentation helps to regulate the policy, not only the convolutional encoder, in DMControl, a previously unknown fact. And we open the door to doing augmentation in the latent image space, improving computational efficiency with a pre-trained encoder.  Of course we also included several ablations of our method to help establish the minimal working algorithm.
>
> This is certainly a spread of ideas, curated with intention.  We aimed to firmly establish that ATC is widely applicable and not dependent on some quirk or gimmick from one type of environment.  And thereafter we point toward some follow-on benefits arising from the flexibility that decoupled representation learning brings to RL, hopefully setting the stage for further investigations by the community.
>
>
> Does this sound reasonable and a subjectively sound contribution?

---

### Official Review · AnonReviewer2 · 2020-10-29
**Official Blind Review**

**Rating:** 6
**Confidence:** 4

**Review:**

This paper proposes an auxiliary task for learning better representations for reinforcement learning. The idea is interesting and is a very active area of research at the moment.

My main concern is that the paper is mostly an empirical evaluation, as the novel algorithm is mostly an extra contrastive loss. Further, it seems their method is almost exclusively meant for pixel-based environments (see point 1 below), so the authors should make this point more explicit. Given this empirical emphasis, I feel the authors could have performed a deeper exploration into understanding _why_ their proposed algorithm performs the way it does in the different environments considered. In particular, some very specific design decisions were made in evaluation (see, for instance, points 2, 5, 9, 11, 12, 13, 14, 15, 16 below).
The clarity of exposition could also use some improvement, as I detail below.

Main questions/concerns:
1. In Section 3, the authors write "This task encourages the learned encoder to extract meaningful elements of the structure of the MDP from observations.". This assumes some type of continuity in pixel space, relative to MDP dynamics, which is in general not true.
2. In Figure 1, why doesn't the momentum encoder go through a residual predictor?
3. It seems $\theta$, $\phi$, $\bar{\theta}$, and $\bar{\phi}$ are all updated independently, is this the case? What is the actual training regimen? Are the ATC and regular RL networks trained concurrently?
4. It would help if you include a proper algorithm in the paper.
5. Above equation (2), the authors say "In our implementation, the positives from all other elements...". It's not clear what "positives from  all other elements" means.
6. In section 4.1 the authors say "multiple seeds were run", please specify how many.
7. In section 4.2 the authors say they are capable of "training the encoder online, fully detached from the RL agent", but how is it fully detached if they share the conv layers?
8. It's not clear what the difference between ATC and UL training is. In some experiments the authors use ATC, in others UL. Are they the same thing? For example, in Figure 3, which ones are ATC? Also in Figure 14 vs Figures 15 and 16?
9. In Figure 3, why does one environment compare with pri and the other with 2x, but not both in both environments?
10. In the **Atari** subsection on the comment of detached training, the authors point out subpar performance on Breakout and SpaceInvaders. On SpaceInvaders it's possible the screen changes could cause issues, but what do the authors think cause the subpar performance in Breakout?
11. In point (iii) of section 4.3, what's the network architecture used for training the RL part?
12. In section 4.3 the authrs say they "drew expert demonstrations from partially-trained RL agents". Were these all drawn from the same checkpoint?
13. In the **DMControl** section, the VAE is trying to reproduce a frame $T$ steps in the future? What is the value of $T$ used? Did you try different values?
14. Similar questin for the **Atari** subsection. Also for this section, does your VAE try to predict individual frames or stacked frames (as frame stacking is common in Atari experiments)?
15. In Figure 8 top, are these after pre-training the encoder? If that is the case, regular RL would have used fewer frames in comparison, no? Where would RL be if left to train for longer?
16. In section 4.5 please clarify what "random shift augmentations" are.
17. In the **Encoder analysis** subsection, what do you mean by "attention"?
18. In Figures 14, 15, and 16 it's not at all clear what we're supposed to be looking for, nor how they show that ATC/UL is focusing on the score/enemy and the others are not.

Minor comments:
1. At the bottom of page 2, the term "POMDP" has not been introduced yet.

---

> ### Author Response · Authors · 2020-11-19
> **Many suggestions/clarifications incorporated // empirical emphasis rather as a strength**
>
> We appreciate your detailed questions, comments, and suggestions!  We clarify each of these below, and have revised the manuscript to incorporate your feedback.  Please see the blue text in the new draft for these revisions.
>
> We agree the the paper is mainly empirical, but we view this as a strength. We (i) show the first successful decoupling of unsupervised representation learning and reinforcement learning that does not suffer in performance relative to end-to-end RL, and (ii) identify the simplest possible algorithm and architecture that achieves strong performance across a diverse range of common pixel-based benchmarks.  We hope that it is rather a strength to include such extensive evaluations (as other reviews have noted), each demonstrating a different setting/aspect for the algorithm performing in reasonable ways, for a fairly comprehensive first assessment.
>
> Deeper probing of why our approach works is an interesting question, and to good extent is included in the ablations and encoder analysis--we’ll try to clarify several points below.
>
> Q1: continuity in pixels relative to MDP dynamics
>
> A1: We think of this as a rather widely-held assumption, that the (subsequent) screen observations contain relevant information to the MDP state and dynamics, as is widely considered for POMDPs.  We can discuss more thoroughly if maybe there is a specific kind of continuity in mind?
>
> Q2: no momentum predictor
>
> A2: As in Grill 2020, the predictor is intended to be a transform from “current” timestep of the anchor to “future” timestep of the positive, and hence is not needed on the positive branch.
>
> Q3: parameter update scheme
>
> A3: The momentum encoder parameters are updated every time the encoder parameters are updated (Table 4, Appendix 3), although a less frequent schedule could probably work.  When the main encoder parameters are updated is flexible and differs by how ATC is used, e.g. online vs offline, so we describe that in the respective experiment sections.
>
> Q4: include a proper algorithm
>
> A4:  Following your suggestion, we included a block Algorithm in Appendix A.1.
>
> Q5: clarity of “positives of all other elements”
>
> A5: Thanks, we have re-ordered the phrasing in the paper.
>
> Q6: number of seeds
>
> A6:  At least 3 seeds for each curve,  added a detailed description to the paper.
>
> Q7: meaning of “fully detached”
>
> A7: We have clarified in the paper that this means no training gradient from the RL loss is used to update the encoder (this term “detached” followed recent previous literature Laskin 2020b).
>
> Q8: ATC vs UL
>
> A8: Thanks, we have unified the text/figures to refer to ATC explicitly in all cases.
>
> Q9: prioritized vs 2x UL training in different environments
>
> A9: This is discussed in the paper--in one environment the agent needs to identify a rarely-seen object associated with reward (so use prioritized), in the other environment it simply benefits from doing more representation learning (it was apparently a bottleneck, so use 2x).
>
> Q10: why subpar on breakout
>
> A10: An interesting question for future research, although we did address it in the offline setting--with certain ablations the ATC encoder focused on the paddle in cases where it needed to represent the ball (Figure 10).
>
> Q11: architecture for policy after offline UL
>
> A11: Same architecture as for online UL, now written in the paper.
>
> Q12: checkpoint for expert data
>
> A12: Each different UL algorithm accessed the exact same data set, now made clear in the paper.
>
> Q13&14: values of T for VAE
>
> A13&14:  These are listed in the appendix: for DMControl we tried 0 and 1, with 1 best; for Atari 0, 1, and 3, with 3 best.
>
> Q15: number of pretraining frames
>
> A15: Yes, also listed in appendix, we used 5e4 transitions from each environment for pre-training.
>
> Q16: meaning of “random shift”
>
> A16: Now defined in the paper.
>
> Q17: meaning of “attention”
>
> A17: Similar to [1], we compute the spatial attention map by mean-pooling the absolute values of the activations along the channel dimension and follow with a 2-dimensional spatial softmax. We clarified this in the revised draft.
>
> [1] Sergey Zagoruyko and Nikos Komodakis. Paying more attention to attention: Improving the performance of convolutional neural networks via attention transfer. In ICLR, 2017.
>
> Q18:  figures 14, 15, 16
>
> A18: Basically, given input observations, spatial attention highlights the where the encoder focuses on. Therefore, Figure 14, 15, and 16 implies that the encoder pre-trained via ATC focuses on the meaningful features, such as the score/enemy, which also explains why ATC encoder is effective for RL.
>
> We hope that these clarifications along with the comprehensiveness of our results will merit your further consideration.  We strove to bring value by showing how widespread are the possible use cases for our method--which all reviewers agree addresses an important area of research--hopefully making it likely to be adopted and to prompt further studies, both empirical and theoretical.

---

> > ### Comment · AnonReviewer2 · 2020-11-24
> > **Thanks**
> >
> > Thank you for your detailed responses, which have clarified most of my doubts. My remaining issue is one that has been raised by others regarding the number of seeds used and the reporting of the variance of the runs. In a response to another reviewer you said you are running more seeds, so I look forward to seeing these before the rebuttal period ends.

---

> > > ### Author Response · Authors · 2020-11-25
> > > **Updated draft with more seeds**
> > >
> > > We've increased the number of seeds for online RL vs ATC experiments from 6 to 10 in DMControl and from 4 to 8 in Atari, without significant change in outcome. (new plots are in the paper Fig 2 and Fig 4) DMLab experiments are underway now, altho they take longer to run.
> > >
> > > Best,
> > > Authors

---

### Public Comment · ~Kuang-Huei_Lee1 · 2020-11-17
**Concerns about the novelty and contribution claims**

I would like to point out that the Augmented Temporal Contrast (ATC) task is very similar to what were proposed in [1] and [2] with only minor differences. The auxiliary setting was done in both [1] and [2]. The decoupled setting has also been done in Sec. H of [2] with similarly good results on DM-Control (I think [4] also decoupled representation learning and it's model-based RL objective). Claiming that “None of these methods attempt to decouple encoder training from the RL loss“ (Sec. 2) is not correct. In Sec. 1, the authors claim that “our main enabling contribution is a new unsupervised task“, but the ATC task itself is not completely new. I don’t think previous contributions are sufficiently discussed and acknowledged, despite [1] is cited.

A minor difference to [1] and [2] is that ATC does not use actions and rewards (which are also ablated in [2] for training on single task). In Sec. 1, the authors claim that “our algorithm is a novel combination of elements that enables generic learning of the structure of MDPs from visual observations, without requiring rewards or actions as input.”. However, a MDP is a 4-tuple (state, action, transition distribution, reward). Not having rewards or actions actually contradicts with the explicit requirements for modeling the complete structure of a MDP. A better description could be something like "Our algorithm captures the temporal coherence of observations in MDPs".

Not having actions has merits in, for example, enabling multi-task encoder training since the action specs of different tasks are not compatible with each other. However, it also has a limitation in off-policy learning, because, instead of modeling s, a -> s’, it models s -> s’ which means you marginalize over action distributions from experience replay instead of online policy. This implications have been discussed in [2] and Sec. 3.4 of [3]. Although, in Sec. 5, the authors stated that "our preliminary efforts to use actions as inputs (into the predictor MLP) with ATC did not immediately yield improvements.", Sec. E of [2] empirically shows that it does lead to degradation on many tasks from DM-control. I think discussing these trade-offs could help the paper become more comprehensive.

Instead of the ATC task itself, it looks like the actual novel part of this paper is in the empirical evaluations of various configurations for training and pre-training representations, e.g. Multi-Task Encoders. The comprehensive experiments and ablations are very interesting. But I have concerns about how the novelty and contributions currently being claimed. I would hope the authors could clarify or modify the part of the main claims.

[1] Schwarzer et al., Data-Efficient Reinforcement Learning with Self-Predictive Representations, arXiv:2007.05929. (NeurIPS 2020)

[2] Lee et al., Predictive Information Accelerates Learning in RL, arXiv:2007.12401. (NeurIPS 2020)

[3] van den Oord et al., Representation Learning with Contrastive Predictive Coding, 2018.

[4] Hafner et al., Dream to control: Learning behaviors by latent imagination. ICLR 2020.

---

> ### Comment · Area_Chair1 · 2020-11-17
> **A Better Positioning of the Submission**
>
> Dear authors,
>
> given the wide spread of the scores in this submission, Kuang-Huei's comments seem very valuable to me in terms of how the submission is positioning itself (Thanks, Kuang-Huei!). In the spirit of providing a better perspective of the field, I'd like to ask you to also address the comments raised here when drafting your response to the reviewers.
>
> By the AC guidelines (https://iclr.cc/Conferences/2021/ACGuide) the two references provided [1, 2] are not considered contemporaneous and should be properly discussed. Nevertheless, I'm also sensitive to the fact that by the time this paper was submitted they were not peer-reviewed conference papers.

---

> ### Author Response · Authors · 2020-11-20
> **comparing decoupled UL to end-to-end RL**
>
> Thank you for taking the time to provide feedback on our manuscript.
>
> And thank you for raising your work [2] to our attention, we have added a citation in our draft to acknowledge it and strengthen our related work discussion. (blue text in the paper)
>
> ---
> **Q1. ATC task is similar to [2]**
>
> **A1.** We agree that there are similarities between our ATC task and your CEB auxiliary task [2], but ATC is more easily understood in relation to the other works we already provide--clearly this is an active area of research we are building upon.  The main motivation for and distinction of the auxiliary task in [2] seems to be compressive ability of the conditional entropy bottleneck (CEB), but we make no use of this mechanism in our loss function (and avoid its additional hyperparameter).  Furthermore, unlike [2], the main point of our paper is to fully decouple UL from RL and compare against end-to-end RL, see Q3/A3.
>
> ---
> **Q2. ATC task is similar to [1]**
>
> **A2.** Although the developments in [1] emphasized spatially local losses, they do include a final experiment which uses a global-only loss similar to ATC (although still not identical), and they provide nice analysis of the MI learning objective, as we have described (and have discussed this reference with the authors).  [1] does not decouple UL from RL.  Although there are certainly closely shared components, it seems to us unlikely that someone reading and reproducing [1] or [2] would arrive at ATC directly (nor its simplicity).
>
> ---
> **Q3. Other methods have decoupled encoder training**
>
> **A3.** It's correct that other methods have tried decoupling UL from RL in their ablations, but they have always taken a performance hit relative to end-to-end---this is true of CURL (Laskin 2020) and appears to be the case for PI-SAC (Fig 2 vs Fig 15 in v2 of [2]). You're also right that world models like Dreamer use a detached encoder. However, Dreamer and similar methods don't show results with the reward gradient backpropagated end-to-end. In our work, we make an apples to apples comparison between the decoupled and end-to-end method, and show that ATC succeeds in decoupling UL from RL without losing performance. We agree that this is nuanced and have clarified this further in the text (blue text) since you're right that other methods have included detached encoder results.
>
> ---
> **Q4. Learning the structure of MDPs from observations**
>
> **A4.** True, we have refined the statement in the introduction to be more precise, that the part of the structure of the MDP being learned is to do with the observations and transitions only.
>
> ---
> **Q5. On not using actions.**
>
> **A5.**  Thank you for pointing out the merits of training ATC without actions.  We were surprised, if a little disappointed, not to discover strong benefits from including the actions either as an input to the ATC model (forward model) nor predicting them as an output (inverse model) in our experiments.  In Section E of [2] (in v2, 26 Oct only), it does not appear to us a clear-cut claim that action-conditioning had a strong effect, as most of the shaded regions in the figure overlap almost entirely.  We found similar, very slight differences (much less than the difference between ATC and other UL methods) and opted for the simplicity of doing without actions.  While it seems intuitive that they should help, perhaps a more clever/meaningful way to introduce them into the model is required, and hopefully by mentioning our attempts in the paper, we give at least a starting clue for others.
>
> ---
> **Q6. Novel empirical contributions**
>
> **A6.** Thank you for acknowledging our experiments, we hope they are a useful reference!

---

### Comment · Area_Chair1 · 2020-11-17
**On the number of seeds used**

Several reviewers raised concerns about the number of seeds used and while the authors are drafting their response I would like to ask for further clarifications surrounding the methodology used to report the results. It is said "Multiple seeds were run for each experiment, and the lightly shaded area around each curve represents the maximum extent of the best and worst seeds." From this excerpt, my questions are:

1. Are the solid lines the average performance across all the seeds?
2. Were the same number of seeds used for each setting?
3. Why is the paper reporting the best and worst seed performance instead of a different summary statistic of the spread of a data distribution (e.g., standard deviation, MSE)? Is the best and worst seed performance more representative to the point being made?

I'm asking these questions to streamline the discussion  about this submission, as I'm sure these questions will inevitably come up.

Thanks in advance.

---

> ### Author Response · Authors · 2020-11-20
> **paper updated with number of seeds**
>
> Q1. solid lines average?
> A1. Yes, exactly, we have updated the draft to state this.
>
> Q2. same number of seeds?
> A2. Within an experiment, each method being compared had the same number of seeds.  Some experiments had more or less seeds than others, mainly based on different time/computation requirements.
>
> Q3. Reporting best/worst vs other summary statistic?
> A3. Some papers report the average with a bold line, and then with faint lines draw all the individual random seeds, but this became too cluttered when we were comparing so many UL methods.  Conveying the maximum extent of the raw results still seemed useful when reporting on a modest number of seeds (3 or 4), especially in the DMControl experiments, some of which can vary widely from run to run (i.e. so if someone is reproducing this and they get some of the high/low runs, it’s straightforward to see if they fall in the range we observed).  Does this seem reasonable?

---

### Decision · Program_Chairs · 2021-01-07
**Final Decision**

**Decision:**

Reject

**Comment:**

This paper introduces ATC, which is a contrastive learning on observations separated in time, to learn representations that do not need to take rewards into consideration. These learned representations allow, for the first time, a real disentanglement between representation learning and control, as the agent can simply load such a representation, “freeze it”, and still recover performance of end-to-end deep reinforcement learning agents.

Overall, all reviewers agree this is a promising direction. Nevertheless, there has been extensive discussion (with the authors and privately) about the significance of the reported results due to the small number of seeds. On one hand, there’s the argument that there is a wide range of experiments and that should compensate for a small number of seeds in individual experiments. On the other hand, there are experiments with as little as two seeds (e.g., DMControl multi-env) and this can be seen at most as anecdotal evidence. There’s also the argument that we, as a community, should be striving for more reliable and meaningful experiments in reinforcement learning. Moreover, there have been concerns about how “variance” is being reported (max and min performance) and, although the authors replied to that, an alternative plotting was never shown.

Importantly, at this point it is not clear how many seeds were used in each experiment (Figures 6, 7, 9, 11, 12, 13 do not report the number of seeds used). It is said that each curve represents a minimum of 3 random seeds, but that is very informal and not that useful. Exactly stating the number of seeds would be the right thing to do, not to mention that in the rebuttal it is said that 8-game pretraining for Atari multi-env uses 2 seeds, contradicting the original claim. Also, sometimes, different methods, in the same experiment, are  “averaged” across different numbers of seeds (“DMLab offline -- ATC is 4 seeds, PC and CPC are 2 seeds each”). This is particularly problematic because of the small number of seeds and potentially high variance. Reporting the max over 4 numbers drawn from a Gaussian distribution is very likely to lead to a larger number than when reporting the max over 2 numbers drawn from the same Gaussian distribution.

I do acknowledge the effort to increase the number of seeds during the rebuttal phase, but it is hard to accept a paper with unknown results. We have very little evidence to believe that going from 2 seeds to 5 seeds is not going to change the results. The reviewers couldn’t agree on the variance of this process as well. Some say the variance of PPO is low between runs when using the same hyper parameters while others mention papers (e.g., Deep RL that matters) that show how much variance one can have across these methods. Thus, I cannot accept this paper conditioned on more seeds being added to the final version because we don’t know what the results will look like. Since this paper is mostly an empirical study, it should have thorough experiments and a careful analysis of the results, but the small number of seeds prevents that in my opinion. Thus, as difficult as it is given the promising direction of the paper, I’m recommending its rejection. I strongly encourage the authors to increase the number of runs in their experiments and to use a more standard measure of variability (e.g., standard error, standard deviation) when reporting their results. This will then be a very strong submission for a future conference.